# The Table Says Otherwise: Testing LLMs with Counterfactual Relational Data

Xinzhi Wang
Purdue University
wang6171@purdue.edu

Chunwei Liu
Purdue University
chunwei@purdue.edu

## ABSTRACT

Large language models (LLMs) are increasingly used to answer natural-language questions over structured data. However, when a table contains familiar real-world facts, it is unclear whether the model answers by reading the provided data or by recalling knowledge learned during pretraining. This distinction is important for database applications, where the provided tables should be the source of truth. In this paper, we introduce CONTRATABLE, a paired original–counterfactual benchmark for evaluating whether LLMs ground their answers in relational tables. We build the benchmark with two aligned versions: an original database with real-world facts and a counterfactual database that preserves the same schemas, identifiers, and relationships while changing selected country, club, and player attributes. We design 214 matched questions across three levels: single-table lookup, multi-table lookup, and multi-table temporal reasoning. Experiments on commercial closed-source and open-source models show that strong instruction-tuned models can often handle direct lookup, but their reliability drops as questions require joins, comparison, and temporal reasoning. The gap between original and counterfactual accuracy reveals that models may fall back on prior knowledge when table evidence conflicts with familiar facts. These results suggest that table-QA evaluation should measure not only accuracy, but also faithfulness to the provided database.

**VLDB Workshop Reference Format:**
Xinzhi Wang and Chunwei Liu. The Table Says Otherwise: Testing LLMs with Counterfactual Relational Data. VLDB 2026 Workshop: NOVAS.

**VLDB Workshop Artifact Availability:**
The source code, data, and/or other artifacts have been made available at https://github.com/AuroraWXZ/LLM_understand_table.

## 1 INTRODUCTION

Large language models (LLMs) are increasingly used as natural-language interfaces to data. Prior work has studied question answering over free-form documents and images [1, 6, 17–20, 22, 27, 29, 31], while recent data-management systems and benchmarks explore how LLMs can help users interact with structured and semi-structured data [5, 7, 10, 15, 26, 32–34, 36? –38]. A simple and flexible approach is to provide a small set of related tables and ask questions in natural language, avoiding the need to write SQL. However, this setting raises a key question: when an LLM

answers a table-based question, does the answer come from the provided tables or from knowledge learned during pretraining [12, 28]? This distinction is important for database applications, where the provided data should be the source of truth. Table values may be private, local, recently updated, or hypothetical, and may intentionally differ from public knowledge in settings such as simulation, what-if analysis, data cleaning, or testing. If an LLM replaces table evidence with familiar facts from pretraining, it may produce an answer that sounds plausible but is wrong with respect to the database. Existing table question-answering benchmarks make this behavior hard to isolate, because many are built from real-world entities that may already appear in pretraining data [5, 7, 13, 15, 26, 38]. As a result, benchmark accuracy can mix two abilities: reasoning over the given tables and recalling world knowledge. A high score may therefore overestimate how faithfully a model follows the provided tables.

To separate these two abilities, we draw inspiration from knowledge-editing research, where counterfactual facts are used to test whether a model follows newly provided information or returns a familiar real-world answer [8, 11, 23, 24]. We adapt this idea from isolated facts to relational databases. Instead of changing a single statement, we construct a counterfactual database that preserves the original schema, identifiers, and relational structure, while replacing selected values with valid conflicting values. This design lets us compare model behavior when the table agrees with real-world knowledge and when it conflicts with it.

In this paper, we introduce CONTRATABLE, a paired original–counterfactual benchmark built from a football transfer-market database. This domain is well suited for our goal: many facts about countries, clubs, and players are likely to be known by LLMs [12, 28], while the database also contains rich relational and temporal structure through players, clubs, countries, competitions, games, appearances, and transfers. The original database contains real-world facts. The counterfactual database keeps the same schemas and relationships, but changes selected attributes such as country capitals, continents, confederations, club countries, domestic competitions, stadiums, player citizenship, birth information, preferred foot, height, and date of birth. The reference answer always follows the provided database. We organize questions into three levels, from single-table lookup to joins and temporal reasoning, to test how grounding changes with reasoning difficulty.

We evaluate both commercial closed-source models and open-source models. Our results show that strong instruction-tuned models can often handle direct lookup questions, even when the table contains counterfactual values. However, as questions require joins, comparison, and temporal reasoning, the gap between original and counterfactual accuracy becomes larger. This counterfactual gap suggests that models become less faithful to the provided database

when the reasoning path is harder and the table conflicts with familiar knowledge. Stronger models and instruction tuning improve overall accuracy, but they do not fully remove this effect.

In summary, this paper makes the following contributions:

- We introduce CONTRATABLE, a paired original–counterfactual benchmark for testing whether LLMs follow tables or prior knowledge.
- We build a consistent counterfactual football database that preserves structure while changing selected real-world facts.
- We design questions spanning lookup, joins, and temporal reasoning to study grounding under increasing complexity.
- We show across commercial and open-source LLMs that table grounding weakens as reasoning paths become more complex.

## 2 RELATED WORKS

Prior work on table question answering evaluates capabilities such as cell lookup, fact verification, numerical operations, and compositional reasoning over structured data. Recent studies directly serialize tables for general-purpose LLMs and show that they can perform competitively on established table-QA benchmarks, particularly with few-shot or chain-of-thought prompting [4]. Tenet uses SQL-based generation and modified tables to create counterfactual training examples for tabular natural language inference and fact verification [3]. While Tenet generates claims from the modified tables for model training, our work treats a counterfactual relational database as the source of truth and tests whether an LLM follows its values when they conflict with knowledge learned during pretraining. This explores counterfactual table modification as an evaluation method for grounding rather than primarily as a mechanism for generating training examples. However, performance on table-QA tasks is sensitive to factors such as table format, row order, prompt design, and table size [30]. Moreover, because these benchmarks commonly contain real-world entities and facts, answer accuracy may conflate reasoning over the supplied table with factual knowledge recalled from pretraining.

Related QA research distinguishes knowledge stored in model parameters from evidence supplied in the input. Longpre et al. [21] show that models may prefer memorized answers when contextual evidence conflicts with their parametric knowledge . DisentQA further uses counterfactual passages to separate answers derived from these two sources [25]. These studies motivate conflict-based evaluation as a test of grounding, but primarily consider unstructured textual contexts rather than relational tables.

Knowledge-editing research provides the closest methodological precedent. ROME introduced CounterFact, which uses counterfactual fact associations to evaluate editing success, generalization, and specificity [23]. CounterFact+ subsequently strengthened the benchmark by dynamically testing unintended effects on related model outputs [14]. Later benchmarks examine more complex consequences: MQuAKE tests whether edited facts propagate through multi-hop questions, while RippleEdits evaluates logical and semantic ripple effects [35]. However, these benchmarks generally evaluate edited model parameters or isolated textual facts. Our work instead keeps the model fixed and modifies an external relational database while preserving its schema and relationships. This paired original–counterfactual design measures whether an LLM

follows table evidence across lookup, joins, aggregation, comparison, and temporal reasoning rather than reverting to memorized world knowledge.

## 3 DATASET CONSTRUCTION

We construct a paired table-QA benchmark from joinable Transfermarkt CSV files covering more than 37,000 players, 400 clubs, 80,000 games, 99,000 transfers, 1.8 million appearances, and 1.1 million game events, together with country and competition data [9]. Focusing on transfers from 2023–2025 and their related entities and events, we create aligned **original** and **counterfactual** databases that share the same schemas, identifiers, and relational structure, while selected facts are replaced with values that conflict with real-world knowledge. Static entity tables provide the modified attributes, whereas transfer and game records support relational and temporal reasoning. As summarized in table 1, the benchmark contains 214 templates: 61 Level 1, 79 Level 2, and 74 Level 3. Each is instantiated once per database, yielding 214 matched question pairs in total.

### 3.1 Question Design

Using the relations available in the dataset, we divide the benchmark into three levels. The level is determined by the database operation required to obtain the answer, rather than only by the number of tables involved. This design connects each natural-language question to a familiar database action and lets us study whether counterfactual grounding changes as the required operation becomes more complex.

Level 1 corresponds to *single-table lookup*. The model only needs to identify one row in one table and read the requested column. Some Level 1 questions include simple row-level computation, such as extracting the year from a date or computing a player's age from date of birth. These questions are similar to a selection followed by projection in a database query. They test whether the model can locate the correct evidence and return the table value, rather than answering from memorized facts.

Level 2 corresponds to *join-based lookup*. The answer cannot be read from a single table, but the relevant entity or event is directly specified in the question. The model must follow one or more join paths, such as connecting a player to their citizenship country, a club to its country, or a club to its domestic competition. The reasoning is still lookup-based: after the joins identify the target row, the answer is obtained by reading an attribute. This level tests whether the model can preserve table evidence across relational links.

Level 3 corresponds to *multi-table temporal reasoning*. The model must first derive which rows are relevant before reading or comparing their attributes across tables. These questions require operations such as joining transfer records with player, club, or country tables, ordering transfers by date, selecting the latest record before a target date, counting rows, comparing values, or aggregating over a set of records. In database terms, this level goes beyond direct lookup and requires joins together with filtering, ordering, grouping, comparison, or interval reasoning.

The increase in difficulty is central to our evaluation. In a direct lookup, the relevant evidence is close to the requested answer. As

**Table 1: The three question levels in our benchmark. Each count is both the number of templates and the number of questions per database, because every template is instantiated once against each database.**

| Level | Definition | Example |
|---|---|---|
| **Level 1: Single-table lookup** (61 questions) | The answer is obtained from one row in one table through selection and projection. A question may also require a simple row-level calculation, such as extracting a year or computing an age. No joins or multi-row reasoning are needed. | *What capital city is listed for {country name}?* |
| **Level 2: Multi-table lookup** (79 questions) | The answer requires one or more joins across tables. The target entity or event is directly identified, so the model follows the join path and reads the requested attribute. No aggregation or temporal interval inference is required. | *What capital city is listed for {player name}'s citizenship country?* |
| **Level 3: Multi-table temporal reasoning** (74 questions) | The model must first derive the relevant rows through joins and operations such as filtering, ordering, aggregation, comparison, or temporal interval reasoning. The answer therefore requires multi-step database reasoning rather than direct lookup. | *On {target date}, was {player name}'s club country on the same continent as the player's citizenship country?* |

joins and intermediate operations are added, the model must carry the provided values through a longer reasoning path. The example questions for each level are shown in table 1. This design allows us to test whether the model continues to rely on the table or falls back on familiar pre-trained knowledge, especially when asked to produce a direct answer. To ensure that this comparison reflects reasoning difficulty, all questions use clear, schema-aligned wording, and every included table contributes to a traceable reasoning path. We next describe how we introduce counterfactual facts while preserving the database's internal consistency.

## 3.2 Counterfactual Design

The counterfactual database preserves the original schema, identifiers, and relational structure while replacing selected real-world facts with valid conflicting values. The evaluated model is not told which database it receives, and the reference answer always follows the provided tables. We modify attributes that are likely to be known by LLMs without changing the database structure. These include country capitals, continents, and confederations; club countries, domestic competitions, and stadiums; and player birth information, citizenship, date of birth, preferred foot, and height.

Internal consistency is central to the design [8, 11]. We preserve the schema, data types, identifiers, primary- and foreign-key relationships, and all join paths used by the questions. Categorical values are reassigned through one-to-one chain swaps over valid values, with no self-mapping or repeated replacement. Semantically coupled fields, such as country names and country codes, are updated together. Numeric and date fields retain their types and receive small valid perturbations, while structural records, including transfers and game events, remain unchanged. With random seed 0, we modify values in the country, club, and player tables. With the original and counterfactual databases aligned, we next generate matched questions that share the same template and reasoning path but derive their answers from the corresponding database.

## 3.3 Question Generation

We first specify the target attributes, join paths, and question level. GPT-5.5 then drafts candidate question templates, which we manually review to refine their wording and reasoning paths and to select entities for the placeholders. For each approved template, a deterministic program computes the reference answer and retrieves the supporting rows. Thus, GPT-5.5 is used only to draft the natural-language wording, while all answers and evidence are

```
You will be answering a table question.
Use only the CSV tables below.
The excerpts preserve the source column order and
selected row order.
Return the final answer plus a brief explanation grounded
in the shown table values.
Use this format: Answer: [short final answer].
Explanation: [brief table-grounded reason].

Question: [QUESTION]

Table: [TABLE NAME] ([CSV FILE])
COLUMNS
ROWS
```

**Figure 1: Single-table user prompt used for direct table question answering. For multi-table questions, the prompt includes all input tables.**

derived programmatically from the database. Each template is instantiated once against the original database and once against the counterfactual database. The resulting pair preserves the same question structure and reasoning path, while its placeholders, reference answer, explanation, and provenance are derived independently from the corresponding database.

For answer generation, we use a zero-shot, end-to-end prompt without chain-of-thought instructions. The system prompt asks the model to produce a concise answer and a brief explanation grounded in the supplied tables. The user prompt contains the question and one or more CSV excerpts, preserving the source column order and the order of the selected rows. The explanation is requested to make the model's use of table evidence observable, particularly for binary questions where a correct *yes* or *no* answer may result from incorrect reasoning. However, the model is not instructed to provide step-by-step reasoning. Figure 1 shows the single-table prompt. For multi-table questions, all input tables are included in the same prompt.

## 4 RESULT

### 4.1 Experimental Overview

We evaluate both commercial and open-source models on CONTRAT-ABLE. The commercial models include Gemini-3.1-Flash-Lite and GPT-5.4-Mini. In addition, we include open-source models because many table-based applications involve private data, where users may prefer to run the model locally instead of sending tables to

an external API. The open-source models include Gemma-4-E2B-it, Gemma-4-E4B-it, Qwen3.5-2B, Qwen3.5-9B, Llama-3.1-8B, Llama-3.1-8B-Instruct, Llama-3.2-3B-Instruct, and Llama-3.2-1B-Instruct. Each question is evaluated under a paired original–counterfactual setting. The question wording and reasoning path stay the same, while the table values may differ between the original and counterfactual databases. This design lets us test whether a model follows the provided table evidence or falls back on its prior knowledge. To keep inputs comparable across questions, we limit the number of rows provided to the model. Ground-truth evidence rows are always included, and the remaining rows are sampled as distractors from the required tables.

## 4.2 Evaluation

For each question, the evaluated model produces a final answer and a brief table-grounded explanation. We score the complete response rather than only the final answer string, since a correct answer, particularly for a binary question, may be supported by reasoning that is inconsistent with the supplied tables. GPT-4o serves as a binary evaluator and receives the question, the model response, and the reference answer with supporting evidence.

To validate this evaluation procedure, we manually inspected GPT-4o's judgments on representative responses across different models, question levels, and database settings. We revised the evaluator prompt multiple times to resolve incorrect or inconsistent judgments, including cases involving alternative answer formats, partially correct responses, and disagreement between the final answer and its explanation. After these revisions, we manually rechecked the evaluation outputs and fixed the prompt for all reported experiments. The same validated procedure is applied to both the original and counterfactual databases.

## 4.3 Results

Table 2 reports model accuracy across the three question levels on both the original and counterfactual databases. Since each counterfactual question keeps the same wording and reasoning path as its original version, the accuracy drop from the original database to the counterfactual database is the main signal we study. We refer to this drop as the counterfactual gap. A larger gap means that the model becomes less reliable when the table evidence conflicts with familiar real-world facts. We first examine how this gap changes as questions move from direct lookup to more complex reasoning.

*Counterfactual gaps reveal reliance on pre-training knowledge.* Level 1 usually has the smallest counterfactual gap because it only requires direct lookup. The gap is nearly zero for strong models: GPT-5.4-Mini has a 0-point gap, and Qwen3.5-9B has a 1.64-point gap. This suggests that when the task is simply to read one cell, models can often follow the provided table even when the value is counterfactual. However, once questions require joins or longer reasoning, the gap becomes much larger. GPT-5.4-Mini increases to a 10.13-point gap at Level 2 and an 18.92-point gap at Level 3. Qwen3.5-9B and Gemma-4-E2B-it also show large Level 2 gaps of 20.25 and 31.65 points, respectively.

This pattern supports our main assumption: as table reasoning becomes harder, models are more likely to fall back on familiar knowledge instead of strictly following the database. The accuracy

drop is therefore not only a difference between two datasets. It also reflects a change in how the model appears to make decisions. We manually inspected the generated explanations for changed predictions and found that, in many cases, the model starts to justify its answer using real-world knowledge rather than the counterfactual values shown in the table. This behavior is especially clear in Level 3, where the model must combine evidence across multiple rows or reason over time. In these cases, the table still contains the required evidence, but the model is less faithful to it when the evidence conflicts with what the model already knows.

The trend is not perfectly monotonic for all models. For weaker models, original accuracy can already be low, leaving less room for a further drop on the counterfactual database. Therefore, the counterfactual gap is most meaningful when the model first performs reasonably well on the original database. Gemini-3.1-Flash-Lite is a small Level 1 outlier, where counterfactual accuracy is slightly higher than original accuracy. We manually checked this case and found that the only original error comes from wording ambiguity rather than counterfactual reasoning: the model finds the correct cell value, but interprets the question as asking whether the literal word "confederation" appears, and answers "No."

*Stronger models help, but do not remove the gap.* We next ask whether stronger models are better at staying faithful to the table when reasoning becomes harder. Level 3 is the most useful setting for this comparison, as it requires multi-table, comparison, or temporal reasoning. Gemini-3.1-Flash-Lite performs best at this level, reaching 94.59% accuracy on the original database and 89.19% on the counterfactual database. GPT-5.4-Mini also performs well on the original Level 3 questions, with 77.03% accuracy, but drops to 58.11% in the counterfactual setting. Among open-source models, Qwen3.5-9B and Gemma-4-E4B-it are the strongest on Level 3, reaching 66.22% and 59.46% on the original database, and 55.41% and 44.59% on the counterfactual database. This shows that stronger models are better at following longer table reasoning paths, but the counterfactual gap remains. The same pattern appears within model families. Moving from Qwen3.5-2B to Qwen3.5-9B improves Level 3 original accuracy from 43.24% to 66.22%, and counterfactual accuracy from 35.14% to 55.41%. Similarly, Gemma-4-E4B-it improves over Gemma-4-E2B-it on both original and counterfactual Level 3 questions. These results support our assumption from another angle: model strength improves table reasoning, but it does not fully prevent the model from being influenced by prior knowledge when the table contains counterfactual facts. We next isolate another factor that may affect this behavior: instruction tuning.

*Instruction tuning matters.* To isolate the effect of instruction tuning, we compare Llama-3.1-8B with Llama-3.1-8B-Instruct. Instruction tuning improves overall original accuracy from 5.14% to 68.22%, showing that the model needs to follow the task instruction and use the provided tables. However, the counterfactual accuracy of Llama-3.1-8B-Instruct is still much lower than its original accuracy. This suggests that instruction tuning helps models read and use tables, but is not sufficient to remove the influence of prior knowledge.

Table 2: Accuracy (%) on the original and counterfactual databases.

| Model | Level 1 | | Level 2 | | Level 3 | | Overall | |
|---|---|---|---|---|---|---|---|---|
| | Original | Counter. | Original | Counter. | Original | Counter. | Original | Counter. |
| Gemini-3.1-Flash-Lite | 98.36 | 100.00 | 94.94 | 88.61 | 94.59 | 89.19 | 95.79 | 92.06 |
| GPT-5.4-Mini | 100.00 | 100.00 | 94.94 | 84.81 | 77.03 | 58.11 | 90.19 | 79.91 |
| Qwen3.5-9B | 98.36 | 96.72 | 89.87 | 69.62 | 66.22 | 55.41 | 84.11 | 72.43 |
| Gemma-4-E4B-it | 96.72 | 98.36 | 92.41 | 72.15 | 59.46 | 44.59 | 82.24 | 70.09 |
| Gemma-4-E2B-it | 91.80 | 86.89 | 74.68 | 43.04 | 45.95 | 33.78 | 69.63 | 52.34 |
| Llama-3.1-8B-Instruct | 88.52 | 68.85 | 78.48 | 35.44 | 40.54 | 31.08 | 68.22 | 43.46 |
| Qwen3.5-2B | 80.33 | 65.57 | 63.29 | 34.18 | 43.24 | 35.14 | 61.21 | 43.46 |
| Llama-3.2-3B-Instruct | 73.77 | 57.38 | 30.38 | 16.46 | 29.73 | 28.38 | 42.52 | 32.24 |
| Llama-3.2-1B-Instruct | 45.90 | 34.43 | 10.13 | 7.59 | 28.38 | 22.97 | 26.64 | 20.56 |
| Llama-3.1-8B | 18.03 | 1.64 | 0.00 | 1.27 | 0.00 | 0.00 | 5.14 | 0.93 |

## 5 FUTURE WORK

We plan to extend ContraTable from a grounding-focused benchmark to a broader evaluation of LLM-based table reasoning and database-oriented methods. In particular, we will compare two paradigms: (i) a text-to-SQL pipeline, where an LLM generates a SQL query that is executed on the database, and (ii) a direct LLM approach, where the model receives the tables and question and produces an answer end-to-end. We will evaluate these systems in terms of accuracy, latency, and inference cost, with the goal of understanding whether end-to-end LLM methods can replace database pipelines or whether each approach is better suited to different workloads.

Counterfactual database construction will remain central in this setting. It enables us to test whether LLMs follow the provided tables or rely on memorized knowledge, while leaving SQL execution unaffected. As a result, it provides a controlled setting to compare the two paradigms, including cases where the database contains values that contradict or are absent from the model's pretraining knowledge.

To support this comparison, we will redesign the benchmark around a structured taxonomy of operations and reasoning complexity. Level 1 will cover single-table operations such as projection, filtering, aggregation, grouping, comparison, and ordering. Level 2 will extend these operations with joins across multiple tables. Level 3 will include two complementary classes of questions. The first consists of queries that are natural for SQL but challenging for LLMs, including long multi-hop reasoning, exhaustive scans over large tables, large intermediate or final result sets, and queries with many interacting conditions that must be preserved during generation. The second focuses on tasks that are difficult to express in standard SQL but more suitable for LLMs, such as semantic matching, semantic joins, and multimodal understanding. This design allows us to evaluate both paradigms under conditions that favor each, rather than treating difficulty as a single axis.

This structured design also enables more fine-grained error analysis. We will introduce a table-free, closed-book baseline to measure whether models possess the relevant knowledge independently of the tables, helping to isolate failures caused by counterfactual conflicts. We will then analyze performance by operation type (e.g.,

joins, comparisons, aggregation, temporal filters) and reasoning characteristics. For counterfactual cases, we will examine whether errors are triggered by modified attributes and whether model outputs align with the original database, allowing us to distinguish knowledge intrusion from general reasoning failures. Extending this analysis to both direct LLM and text-to-SQL systems will provide a clearer picture of where each paradigm succeeds, where it fails, and how they can complement each other in database-centric applications.

## 6 CONCLUSION

This paper studies whether LLMs answer table-based questions by following the provided database or relying on prior knowledge. We introduce ContraTable, a paired original–counterfactual benchmark for testing whether LLMs follow relational tables or rely on prior knowledge. By preserving the schema and relations while changing selected facts, the benchmark directly measures table grounding under knowledge conflict.

Our experiments show that the counterfactual gap grows from direct lookup to joins and temporal reasoning. Stronger models and instruction tuning improve accuracy but do not ensure faithful use of the supplied data. Future work will extend the benchmark with more systematic operation coverage, stronger baselines and error analysis, and a comparison between direct LLM answering and text-to-SQL pipelines. This broader evaluation will help determine when LLM-based methods can replace, complement, or should defer to traditional database systems.

## ACKNOWLEDGMENTS

This work used Anvil at Purdue through allocation CIS251032 from the Advanced Cyberinfrastructure Coordination Ecosystem: Services Support (ACCESS) program [2], which is supported by U.S. National Science Foundation grants 2138259, 2138286, 2138307, 2137603, and 2138296. This material is based upon work supported by the Google Cloud Research Credits program with the award EDU Credit-wilsonjessica- 490123310. Some results presented were obtained using the Chameleon testbed [16] supported by the NSF.

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
