# OpenReview forum: "The Table Says Otherwise: Testing LLMs with Counterfactual Relational Data"
_VLDB.org/2026/Workshop/NOVAS — NOVAS 2026_

### Official Review · Reviewer_SkVh · 2026-07-11

**Confidence:** 4

**Improvement Opportunities:**

O1. Adding some extra analysis and information to understand better the failure modes is beneficial:
- Are we certain the failures are exactly for the queries with changed input? If not for 100% of them, what percentage is caused by the counterfactuals?
- For the failed queries on the counterfactual database, does the LLM answer correctly the query when taking the original database into account or is it wrong for both?

O2. Given the work is primarily experimental, I would urge the authors to also include the prompt structure if possible. Is CoT used? Do you use few-shot examples?

**Minor Comments:**

NA

**Short Summary:**

The authors propose an evaluation technique to see if LLMs are likely to use the provided data in context as opposed to that in its weight making an argument for distinguishing between reasoning over the given tables and recalling world knowledge.

**Strong Points:**

S1. Interesting problem as it relates to updating data in context instead of weight updates and seeing which of the two model is likely to rely on for answering.

S2. Strong results indicating that LLMs are more likely to rely on world knowledge instead of provided input data.

S3. The experimental evaluation aims to propose solutions: more capable models and instruction tuning.

---

### Official Review · Reviewer_KdZ6 · 2026-07-13

**Confidence:** 4

**Improvement Opportunities:**

**(O1)** It would be interesting to test if the model does indeed know the information that is being replaced with the authors' intervention. Specifically, perhaps it would be possible to let the model answer the question without having any data available, to see how well it would perform while only relying on learned information. I know that this doesn't answer the specific question targeted by the paper, but it could still potentially be a useful signal to interpret the behavior of different models.

**(O2)** GPT-4o is used as a checker of answers, which, to my understanding, is relatively standard practice. However, like any LLM, it is prone to hallucinations, so I would still prefer to see some evidence of its correctness or some mechanism to validate its outputs.

**(O3)** The authors make an effort to investigate the origin of model errors, which is great. However, currently the investigation is only over hand-picked example cases. It would be good if they could develop some methodology to systematically and quantitatively evaluate these errors, perhaps by classifying them and reporting the frequency of different types of errors.

**Minor Comments:**

**(D1)** Since this is a workshop paper, and presumably, the authors would like to develop this into a full conference paper. It would have been good to include a set of future plans demonstrating their vision for a full comprehensive study, which could be presented at the workshop and represent an opportunity to gather feedback.

**Short Summary:**

The authors present a benchmark for LLM question answering over structured data that tests whether the LLM relies on the information present in the provided data or on the information learned during pretraining. To achieve this, the authors build a tabular dataset about football teams and optionally apply an intervention where they replace some true facts with values that are factually false but can be useful to test whether the model is making answers based on the data alone. The benchmark is made up of three levels of questions with increasing difficulty and need for LLM reasoning: single table lookup, multi-table lookup, and multi-table temporal reasoning (i.e., questions that require filtering, aggregation, and reasoning about the results). The empirical results are generated over several open-weight and proprietary models. They demonstrate a persistent accuracy gap between the data with and without the intervention, indicating that all models rely on learned knowledge to some extent, especially for the hardest questions that involve reasoning.

**Strong Points:**

**(S1)** The question being addressed by this paper is pretty interesting, and the presented benchmark can be a useful target task when developing future LLMs.

**(S2)** The methodology is very simple but represents a neat and effective idea.

**(S3)** The paper is overall well-structured and easy to follow, with sufficient context and examples.

---

### Official Review · Reviewer_VbFh · 2026-07-16

**Confidence:** 4

**Improvement Opportunities:**

O1.
The use of perturbed relations is not fully new. Tenet (https://dl.acm.org/doi/10.1145/3626730) modifies tables to generate counterfactual examples for tabular fact checking.
However, Tenet uses the modified table to generate claims, while ContraTable treats the modified table as the source of truth and tests whether the model follows it with conflicting prior knowledge.
I suggest to discuss this connection explicitly to strengthen the paper by mentioning fact-checking example generation as another possible use of the proposed method.

O2.
A drop in accuracy does not always mean that the model used prior knowledge. It may also reflect ordinary reasoning errors or unnatural counterfactual combinations.
A useful analysis would report how often a model that fails on the counterfactual version gives exactly the answer from the original database.
Also, a closed-book baseline would help. If the model cannot answer the original question without the tables, then prior knowledge is less likely to explain the error.

O3.
Paper should report the number of templates, how many examples come from each template, and which integrity constraints are enforced when values are changed.

O4.
Results by operator, such as joins, comparisons, counts, ordering, temporal filters, would help explain where the failures come from.

**Minor Comments:**

n.a.

**Short Summary:**

This paper presents a benchmark for testing whether LLMs follow the data given in relational tables or rely on facts learned during pretraining.
The benchmark includes original and counterfactual versions of a football database. The schema and relationships stay the same, while selected facts are changed. The authors evaluate 214 paired questions across three levels: single-table lookup, multi-table lookup, and temporal reasoning.
The results show a clear drop in accuracy on counterfactual data, especially for harder questions. This suggests that models do not always treat the provided database as the source of truth.
Overall, the problem is relevant and the benchmark useful: simple but effective

**Strong Points:**

S1.
For database applications, the provided data should be authoritative, even when it conflicts with public knowledge. The paper targets this issue.

S2.
The paired original and counterfactual databases make the results easy to interpret.
The split between direct lookup, joins, and temporal reasoning helps show where grounding starts to fail.

S3.
The same construction could also be used to generate examples for tabular fact checking.

---

### Decision · Program_Chairs · 2026-07-16

**Decision:**

Accept

**Comment:**

ContraTable introduces a simple and effective benchmark for testing whether LLMs treat provided relational data as authoritative when it conflicts with memorized world knowledge. The paired original and counterfactual databases, combined with multiple reasoning levels and broad model coverage, produce clear and useful findings. We hope this benchmark sparks productive discussion on grounding, reasoning, and evaluation at the workshop.